

# *LoTo*: a graphlet based method for the comparison of local topology between gene regulatory networks

Alberto J. Martin[1,2], Sebastián Contreras-Riquelme[1,3], Calixto Dominguez[1] and Tomas Perez-Acle[1,2]

[1] Computational Biology Laboratory (DLab), Fundacion Ciencia y Vida, Santiago, Chile
[2] Centro Interdisciplinario de Neurociencia de Valparaíso, Valparaiso, Chile
[3] Facultad de Ciencias Biologicas, Universidad Andres Bello, Santiago, Chile

## ABSTRACT

One of the main challenges of the post-genomic era is the understanding of how gene expression is controlled. Changes in gene expression lay behind diverse biological phenomena such as development, disease and the adaptation to different environmental conditions. Despite the availability of well-established methods to identify these changes, tools to discern how gene regulation is orchestrated are still required. The regulation of gene expression is usually depicted as a Gene Regulatory Network (GRN) where changes in the network structure (i.e., network topology) represent adjustments of gene regulation. Like other networks, GRNs are composed of basic building blocks; small induced subgraphs called graphlets. Here we present *LoTo*, a novel method that using Graphlet Based Metrics (GBMs) identifies topological variations between different states of a GRN. Under our approach, different states of a GRN are analyzed to determine the types of graphlet formed by all triplets of nodes in the network. Subsequently, graphlets occurring in a state of the network are compared to those formed by the same three nodes in another version of the network. Once the comparisons are performed, *LoTo* applies metrics from binary classification problems calculated on the existence and absence of graphlets to assess the topological similarity between both network states. Experiments performed on randomized networks demonstrate that GBMs are more sensitive to topological variation than the same metrics calculated on single edges. Additional comparisons with other common metrics demonstrate that our GBMs are capable to identify nodes whose local topology changes between different states of the network. Notably, due to the explicit use of graphlets, *LoTo* captures topological variations that are disregarded by other approaches. *LoTo* is freely available as an online web server at http://dlab.cl/loto.

# INTRODUCTION

In biological sciences, networks are becoming one of the main tools to study complex systems (*Newman, 2010*). Networks are employed to represent metabolic pathways (*Palumbo et al., 2005*), signaling cascades (*Pescini et al., 2012*; *Ben Hassen, Masmoudi & Rebai, 2008*), and protein-protein interactions (*Wuchty, Oltvai & Barabási,*

Corresponding author
Alberto J. Martin, ajmm@dlab.cl, proteinomano@gmail.com

*2003*), among others. Networks used to represent the regulation of gene expression are known as Gene Regulatory Networks (GRNs) (*Hu, Killion & Iyer, 2007*; *Rodríguez-Caso, Corominas-Murtra & Solé, 2009*). GRNs are directed networks where nodes represent genes, and the links between nodes exist solely if the regulatory element, e.g., a Transcription Factor (TF), encoded by a *source* gene directly regulates the expression of another *target* gene. Major applications of GRNs are intended to perform differential studies in which diverse states of a network representing the same biological system are compared (*Davidson et al., 2002*; *Shiozaki et al., 2011*; *Yang & Wu, 2012*; *Cheng, Sun & Socolar, 2013*; *Gaiteri et al., 2014*; *Okawa et al., 2015*). Interestingly, the structural similarity between two networks can be established at various levels, ranging from the comparison of global network properties to the identification of single nodes and edges whose relationship with the rest of network elements varies. Network properties that can be used to compare networks and therefore to asses their structural difference include the distribution of connections versus non-connections (density), diameter, size/order, connectedness, betweenness, centrality and the distribution of node degree (*Newman, 2010*).

Networks are composed of small induced subgraphs called *graphlets* (*Przulj, Corneil & Jurisica, 2004*). Graphlets represent structural patterns of networks that in the case of GRNs, may encode diverse functional and biologically relevant roles (*Knabe, Nehaniv & Schilstra, 2008*). Statistically over-represented graphlets are usually called *motifs* (*Milo et al., 2002*), but over-representation depends on the null model employed as baseline (*Artzy-Randrup et al., 2004*; *Przulj, Corneil & Jurisica, 2004*). Moreover, the existence of some graphlets has been functionally characterized in GRNs of different organisms, ranging from bacteria to higher animals (*Shen-Orr et al., 2002*; *Ronen et al., 2002*; *Odom et al., 2004*; *Zaslaver et al., 2004*; *Levine & Davidson, 2005*; *Alon, 2007*; *Boyle et al., 2014*). Graphlets can be characterized by the number of their component edges and nodes, are classified accordingly. The smallest graphlets occurring in directed networks are composed of two nodes, while those most frequently employed to characterize networks are graphlets composed of three nodes (*Milo et al., 2002*). Despite larger graphlets constituted of $n$ nodes can be described and used to characterize networks, all of them can be decomposed into at least one graphlet formed by $n-1$ nodes (*Aparício, Ribeiro & Silva, 2015*). In addition, the use of larger graphlets is limited by the computational cost of their enumeration which, depending on the network, could be highly expensive (*Tran et al., 2015*). As expected, several Graphlet Based Metrics (GBMs) can be employed to characterize and compare networks (*Yaveroğlu, Milenković & Pržulj, 2015*). These include graphlet distribution (*Przulj, Corneil & Jurisica, 2004*; *Sporns & Kötter, 2004*), graphlet degree distribution (*Przulj, 2007*; *Koschützki & Schreiber, 2008*; *McDonnell et al., 2014*), graphlet correlation distance (*Yaveroğlu et al., 2014*) and graphlet reconstruction rate (*Martin et al., 2016*). Nevertheless, with the exception of graphlet reconstruction rate, all these GBMs describe global properties of networks disregarding local differences that could be important to compare different states of biological networks. Therefore, in this work GBMs are proposed to describe and compare the properties of diverse states of a network and for instance, to identify the elements that differ between states.
**Table 1 Description of graphlet types.** The number of required TF-coding genes, true edges, false edges is shown for each graphlet type.

| Graphlet type | 1 | 2 | 3 | 4 | 5 | 6 | 7 | 8 | 9 | 10 | 11 | 12 | 13 |
|---|---|---|---|---|---|---|---|---|---|---|---|---|---|
| TF required | 1 | 2 | 2 | 2 | 2 | 2 | 3 | 3 | 3 | 3 | 3 | 3 | 3 |
| True edges | 2 | 2 | 3 | 2 | 3 | 4 | 3 | 4 | 3 | 4 | 4 | 5 | 6 |
| False edges | 4 | 4 | 3 | 4 | 3 | 2 | 3 | 2 | 3 | 2 | 2 | 1 | 0 |

This study describes *LoTo*, an online web-server for the comparison of different states of a GRN. *LoTo* treats the existence or absence of graphlets in two compared networks as a binary classification problem (*Baldi et al., 2000*; *Davis & Goadrich, 2006*; *Powers, 2011*). To do so, *LoTo* assigns a type of graphlet to each triplet of nodes in the two compared network states. This step is done with an efficient method that takes advantage of the sparsity of GRNs: the majority of edges are false or nonexistent, and they originate from the fraction of nodes representing regulator-coding genes. Next graphlet types assigned to the same triplet of nodes in both network states are compared via the construction of confusion matrices. In the final step, the topological similarity between the two networks is quantified by calculating several metrics from these confusion matrices. In this way, *LoTo* first performs a comparison of the global topology; and then it identifies variations in the local topology of each node. Interestingly, the approach implemented in *LoTo* is able to capture topological variations that are not detected by other metrics and would be disregarded otherwise.

In this work, we first propose our definition of graphlets to later explain the GBMs employed in *LoTo*. We then demonstrate how GBMs are more sensitive to random edge removal than their single edge counterparts. We also provide an example where we compare two condition specific states of the *Escherichia coli* GRN: a knock-out of *ompR*, a TF that controls the expression of genes involved in the response to osmotic and acid stresses (*Stincone et al., 2011*), with the control condition. This comparison also emphasizes that TF-coding genes whose local topology changes according to our GBMs are different to those detected by other metrics. Hence, we propose *LoTo* as a novel tool to identify changes in the local structure of GRNs.

## METHODS

### Expanding the definition of graphlets

In this study, graphlets are defined as small induced subgraphs formed by three nodes with at least two regulatory relationships (true edges) between them. Thus, considering all possible connectivity patterns that meet the previous definition, 13 graphlets could be formed (Fig. 1). Importantly, the classical definition of graphlets proposed in *Milo et al. (2002)* was expanded by making both the presence and absence of edges between nodes, equally relevant. Under this definition, all graphlets depicted in Fig. 1, except number 13, require non-existing regulatory relationships (false edges) between nodes (see Table 1).
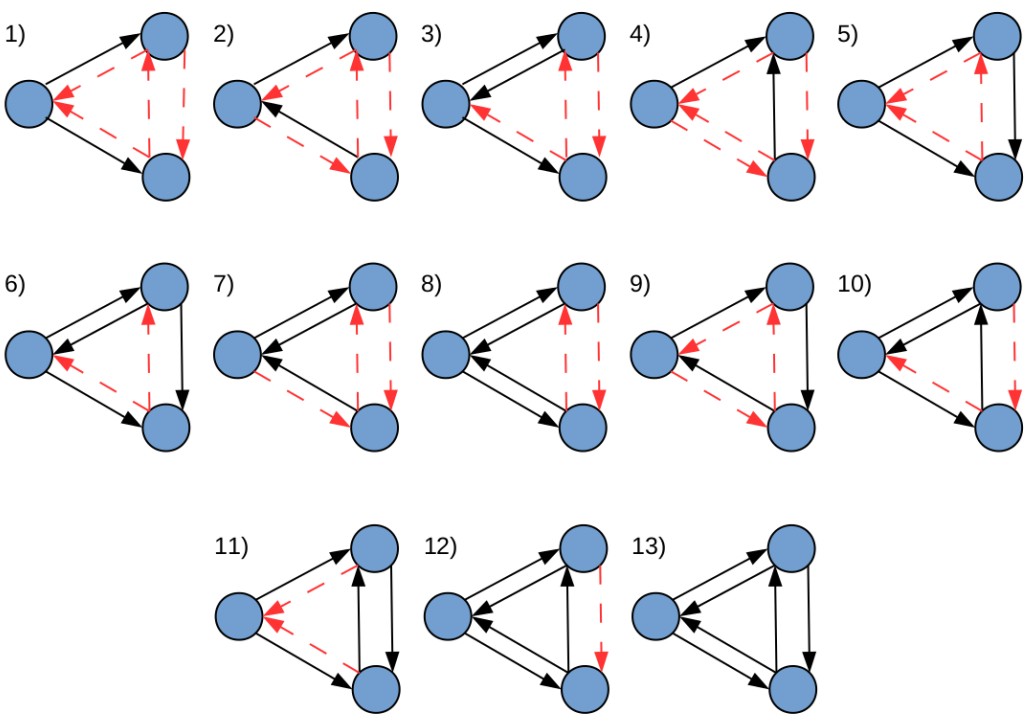

**Figure 1  All possible realizations of three node graphlets that can be defined in *LoTo*.** The direction of edges indicate the sense of the transcriptional regulation. Black edges denote true interactions, and red-dashed edges depict false ones. In this definition, true and false edges are given equal relevance. Adapted from *Milo et al. (2002)*.

## Comparing the structure of GRNs

Let $G$ be a state of a GRN with $V$ nodes and $E$ edges, we want to compare its topology with another state of the same network $G'$. $G'$ should be composed of a set of nodes $V'$, at least partially shared with $G$, and a set of edges $E'$. Thus, one should perform a comparison between the local topology of $G = (V, E)$ and $G' = (V', E')$.

### Similarity metrics derived from graphlet based confusion matrices

As mentioned before, the problem of enumerating the occurrence of graphlets in two networks is treated as a binary classification problem. By doing so, graphlet or node specific confusion matrices are built. A confusion matrix or contingency table, is a table in which each column contains the occurrence of predicted instances and each row shows the actual class of those instances. Therefore, the confusion matrix contains the number of correctly and incorrectly classified true and false examples grouped into True Positives (TPs), False Positives (FPs), True Negatives (TNs) and False Negatives (FNs). Hence, TPs are graphlets present in the two networks; FPs are graphlets found in $G'$ but absent in $G$; FNs are graphlets found in $G$ but absent in $G'$; and TNs are graphlets absent in both network states. It is important to clarify that even if the terminology employed to define the elements of the confusion matrix seems to indicate that one of the compared states is correct and the other is incorrect, this is not the case in *LoTo*. In binary classification problems, the goal is to establish how similar are the predictions of a test set to that of the

actual known classes, i.e., how many of the predictions are correct. In consequenze, a TP example is a true example that was correctly predicted, a FP is a true example misclassified as false and so on. In the comparison of network states performed by *LoTo*, there is not such a thing as the actual class (or type) of a graphlet, and thus, TP graphlets are found in the two states, TNs are absent in both states and FPs and FNs are graphlets present only in one of the two states and absent in the other one. Importantly, the confusion matrices can be built in three different fashions depending on which graphlets are used in their construction and on the purpose of the comparison. If one wants to compare the local topology of single nodes, only the graphlets in which each node participates in $G$ and $G'$ are used; if the purpose is to compare the overall local topology of the two networks, then the matrix can be constructed employing all graphlets in both networks; and finally, confusion matrices can be built with only certain types of graphlets, e.g., all graphlets of type 13. In this work we only focus on the application of GBMs to identify variations in the local topology of single nodes or node-based GBMs and on the comparison of overall network topology or global GBMs.

Several performance metrics can be calculated from a confusion matrix (*Baldi et al., 2000*). *LoTo* focuses on those commonly used to evaluate binary classifiers; Recall (R, Eq. (1)), Precision (P, Eq. (2)), their harmonic mean F1 (Eq. (3)), and Mathews Correlation Coefficient (MCC, Eq. (4)). It is also very important to clarify that both MCC and F1 are symmetric, i.e., their values do not depend of which network state is used as reference to determine FP and FN graphlets, since their values are the same either way.

- Recall:
$$R = \frac{TP}{TP + FN}; \tag{1}$$

- Precision:
$$P = \frac{TP}{TP + FP}; \tag{2}$$

- F1 score:
$$F1 = \frac{2PR}{P + R}; \tag{3}$$

- Matthews Correlation Coefficient (MCC):
$$MCC = \frac{TP \times TN - FP \times FN}{\sqrt{(TP + FP)(TP + FN)(TN + FP)(TN + FN)}}. \tag{4}$$

### Comparison of GBMs and single-edge based metrics as metrics of global network similarity

GBMs and their single edge counterparts were compared on a reference network to determine their sensitivity to variations in a controlled environment. To do so, RegulonDB (*Salgado et al., 2013*) version 8.7 was used to construct a gold standard or reference GRN of *E. coli*. All TF-coding and all non-TF-coding genes with at least one regulatory interaction in RegulonDB were kept. Notably, RegulonDB only contains information about true edges,
actual regulatory interactions, therefore, false edges were assumed to occur between nodes that are not linked.

In order to establish a fair comparison between single-edge based metrics and GBMs, the *E. coli* gold standard network was randomized in two different ways. First, randomly chosen true connections were removed by transforming them into false edges. This procedure is termed *REMO* hereinafter. Second, randomly selected true connections were transformed into false edges, and for each true edge that was transformed, a randomly selected false edge was transformed into a true edge. Hence, the randomized network maintains the same number of true edges as in the original network but the distribution of node degree varies with the changes. This second procedure is termed *SWAP* hereinafter. The two randomization procedures were repeated varying the percentage of changed edges from 0% to 100%. In REMO, removed true edges were transformed into FN edges. On the other hand, in SWAP, removed links were transformed into FN edges and removed false edges were transformed into FP edges. These randomizations were intended to evaluate the behavior of the metrics using a dataset for which the actual percentage of change produced by random alterations is known. To reduce possible dependences on the randomization and to allow proper statistical comparisons, both protocols were repeated $1 \times 10^3$ times, each with a different seed for the random number generator.

*Estimation of the contribution of each graphlet type to GBMs.* Confusion matrices built for every graphlet type were used to determine the relevance of each type in the calculation of global GBMs. To do so, F1 and MCC were calculated and averaged for the thirteen types on the $10^3$ replicas of both *REMO* and *SWAP* at each percentage of randomization. Averaged values were added to then calculate the proportion over the total sum of the metric for all types at every percentage of randomization. On top of helping to determine which graphlet type dominate the metrics, this analysis also allowed to study how the different types fluctuate over the randomization procedures.

### Comparison of GBMs with node centrality differences to identify nodes whose local topology varies

To further validate if GBMs calculated for single node confusion matrices, i.e., node based GBMs, implemented in *LoTo* are related to other methodologies, they were compared to a more traditional approach considering differences in node centrality metrics. Node centralities were computed for all TF-coding nodes in Cytoscape version 3.3.0 (*Shannon et al., 2003*) in two condition specific GRNs of *E. coli* whose construction is described below.

NetworkAnalyzer (*Assenov et al., 2008*), a built-in tool of Cytoscape, was employed to calculate the following centrality metrics: Average Shortest Path Length, Betweenness Centrality, Closeness Centrality, Clustering Coefficient, Eccentricity, Degree, Indegree, Outdegree, Stress Centrality and Neighborhood Connectivity, see *Newman (2010)* and *Assenov et al. (2008)* for their definitions. Pearson's and Spearman's correlations were calculated between GBMs and the differences in node centralities to discern if there is a relationship between them. Correlation coefficients were calculated using the R package version 3.0.2 (*R Core Team, 2013*). *P*-values provided by R were utilized to determine the significance of the correlation coefficients (*p*-value $\leq 0.01$).

*Construction of condition specific networks from gene expression data.* A comparison between two condition specific networks that represent *E. coli* in two different states is used as an application example of *LoTo*. These two networks or states of the *E. coli* GRN were built following a similar approach to *Faisal & Milenković (2014)*, where protein-protein interaction networks were constructed using gene expression micro-arrays. Instead of considering interactions between proteins whose coding genes were expressed in a micro-array, here only known regulations from TF-coding genes whose expression was detected were maintained. These regulations are kept independently of the presence or absence of the target gene. In this way, gene expression data for *E. coli* previously used to study resistance to acidic environments in *Johnson et al. (2014)* was employed to generate the condition specific networks. Four different *E. coli* RNA profiles, each with two replicas, were reported in *Johnson et al. (2014)*, but for the sake of simplicity, we only employed, analyzed and compared two of them, the wild-type and the knock-out *ompR*, a TF that controls the expression of genes involved in the response to osmotic and acid stresses (*Stincone et al., 2011*). Since there are two different replicas of each experiment, regulator-coding genes were considered as expressed if at least one of their specific probes showed a significant signal in each of the replicas (author reported $p$-values $< 0.05$).

*Functional characterization.* Genes regulated by TF-coding genes which were absent in one of the two network states were characterized by manually querying RegulonDB (*Salgado et al., 2013*) and EcoCyc (*Keseler et al., 2010*).

### Algorithm for graphlet enumeration

*LoTo* uses an efficient algorithm to enumerate graphlets in directed networks similar to other graphlet enumeration algorithms previously published (*Wernicke, 2005*; *Aparício, Ribeiro & Silva, 2015*; *Tran et al., 2015*). Since graphlets involve three nodes, a brute force implementation would have a complexity of $O(n^3)$, where $n$ is the total number of nodes in the network. In GRNs, edges only connect regulator-coding genes to their targets, therefore, one can reduce the complexity to find graphlets to $O(t * n^2)$, where $t$ is the number of regulator-coding genes. In our implementation, networks are represented using an adjacency list. The adjacency list contains only true edges arising from regulator-coding genes, thus, allowing to take advantage of GRNs being sparse and edges originating only from a fraction of the nodes. Self-connections are not included in the adjacency list, so the three nodes forming a graphlet are forced to represent different genes. For each regulator-coding gene, a loop over each of its true connections stored in the adjacency list is carried out. This reduces the computational cost in finding the first true edge of each graphlet from $O(t * n)$ to $O(t * k)$, where $t$ is the number of regulator-coding genes and $k$ is the number of their outgoing true connections. Therefore, the total estimation of computational complexity of the algorithm to find graphlets becomes $O(t * k * n)$, where $k$ is at most an order of magnitude smaller than $n$ in real whole genome GRNs.

### LoTo Web server

The web-server allows to characterize a single network, reporting the occurrence of each graphlet type in it, or to perform a comparison between two states of a network. For the

latter, the user needs to provide two directed networks: one used as reference network, and a second network that will be compared to the first. Instead of binary values to define the type of edge, the true connections can be established with a number in the $[0, 1]$ range provided as an edge weight. This number can be used to represent a score or $p$-value of each true edge. False edges are defined as those with an edge weight below a user-defined threshold and edges found in the reference network that are not explicitly defined in the second network. Importantly, *LoTo* accepts several network file formats of common use (tsv, sif, xgmml, cyjs, graphml and gml).

The output page of the web server shows a table in which both single-edge and GBMs are displayed. The metrics included in the table are those described above, plus two metrics named REC and REC Graphlet Degree or RGD that are based on the rate of graphlet reconstruction (*Martin et al., 2016*). REC measures how many of the edges, both true and false, present in a graphlet found in a network state are also present between the same nodes in the second state, and RGD is the average REC for all graphlets in which the same node participates. The web server also generates an output file containing several more GBM metrics and tables describing the comparison. This file also shows the number of graphlets in which regulator-coding and non-regulator-coding genes participate, listing each graphlet that is accounted as TP (present in both network files), FN (only present in the reference network) and FP (only present in the second network). By looking at the lists of FNs and FPs, one can identify the subnetworks formed by nodes whose local topology varies between the two compared networks, and thus might show different regulation.

*LoTo* also produces several additional output files, including a xgmml file containing a network where different colors are used to visualize variations in the compared networks in Cytoscape; together with two other files containing a table describing edges and nodes. For more information and a more detailed description of both the input and output files, please visit http://www.dlab.cl/loto.

## RESULTS

### Graphlet characterization of GRN
#### *Characterization of the RegulonDB gold standard*
Starting from RegulonDB version 8.7, a gold standard GRN was built (see Methods). This GRN is formed by 1,805 genes, of which 202 encode for TFs, and 4,511 true edges. As expected, the number of false edges is much higher than that of true edges, surpassing more than $3 \times 10^6$. The occurrence of each graphlet type found by *LoTo* in this GRN is shown in Table 2. Interestingly, only 11 nodes are isolated and do not participate in any graphlet.

#### *Characterization of condition specific GRNs*
Table 3 characterizes the two network states that represent gene expression regulation for wild-type *E. coli* and a knock-out of *ompR*. As shown, the occurrence of TF-coding genes, the total number of genes and the number of connections between them is slightly smaller than in the gold standard. This decrease in network components is caused by the procedure followed in their construction, i.e., some genes in the gold standard were not present in the

**Table 2  Graphlets occurrence in the condition specific GRNs and in the reference network.**

| Graphlet type | 1 | 2 | 3 | 4 | 5 | 6 | 7 | 8 | 9 | 10 | 11 | 12 | 13 |
|---|---|---|---|---|---|---|---|---|---|---|---|---|---|
| Reference | 329819 | 6305 | 1634 | 4338 | 1641 | 488 | 89 | 5 | 0 | 8 | 31 | 3 | 1 |
| Wild-type | 329790 | 6302 | 1634 | 4307 | 1578 | 488 | 89 | 5 | 0 | 8 | 31 | 3 | 1 |
| *ompR* | 329685 | 6060 | 1592 | 4154 | 1552 | 485 | 82 | 3 | 0 | 6 | 27 | 3 | 1 |

**Table 3  Characterization of condition specific GRNs of *E. coli*.** The number of TF-coding genes (TF), total number of genes (V), existing regulations (EP) and the number of nodes that do not participate in any graphlet (NG) for the two GRNs representing wild-type *E. coli* and the *ompR* knock-out.

| GRN | TF | V | EP | NG |
|---|---|---|---|---|
| Wild-type | 196 | 1796 | 4478 | 11 |
| *ompR* | 189 | 1787 | 4437 | 11 |

transcriptomic experiments or they were not expressed. The occurrence of each graphlet type in these two networks is shown in Table 2. Following the same tendency observed with nodes and edges, and for the same reasons, graphlets are also slightly less frequent than in the gold standard network.

## Comparison of GBMs with single-edge based metrics on the randomized gold standard

We assessed the sensitivity of GBMs and single-edge based metrics on two types of randomization of the *E. coli* reference network. To do so, F1 and MCC were calculated considering both graphlets and single edges on $10^3$ replicas of SWAP and REMO randomizations. The averaged metrics calculated for all replicas are shown in Fig. 2. As seen in all four panels, according to the same percentage of random changes, both metrics calculated for graphlets lay below single-edge metrics. Standard deviations for averaged F1 and MCC are not shown in Fig. 2, since they overlap the averaged metric lines. We also studied the contribution of each graphlet type to the graphlet based versions of F1 and MCC (Fig. 3). In this case, both randomization procedures behave in a similar way, as the percentage of randomization increases the occurrence of simpler graphlets, i.e., types 1 to 6, becomes predominant and thus, they dominate the metrics. On the other hand, graphlets that require their three nodes to be regulator-coding genes, i.e., more complex graphlets, are only relevant at lower percentages of randomization since at higher randomization they are only present in the reference network.

## Comparison of node-based GBMs with differences in node centralities in the comparisons of the condition specific GRNs

With respect to comparisons of node-based GBMs and differences in node centralities, Table 4, Pearson's and Spearman's correlations were calculated between all metrics for all TF-coding genes in the comparison of condition specific GRNs. Interestingly, both coefficients indicate better correlation when calculated between the differences than when they were calculated between the differences and GBMs. This tendency is more evident

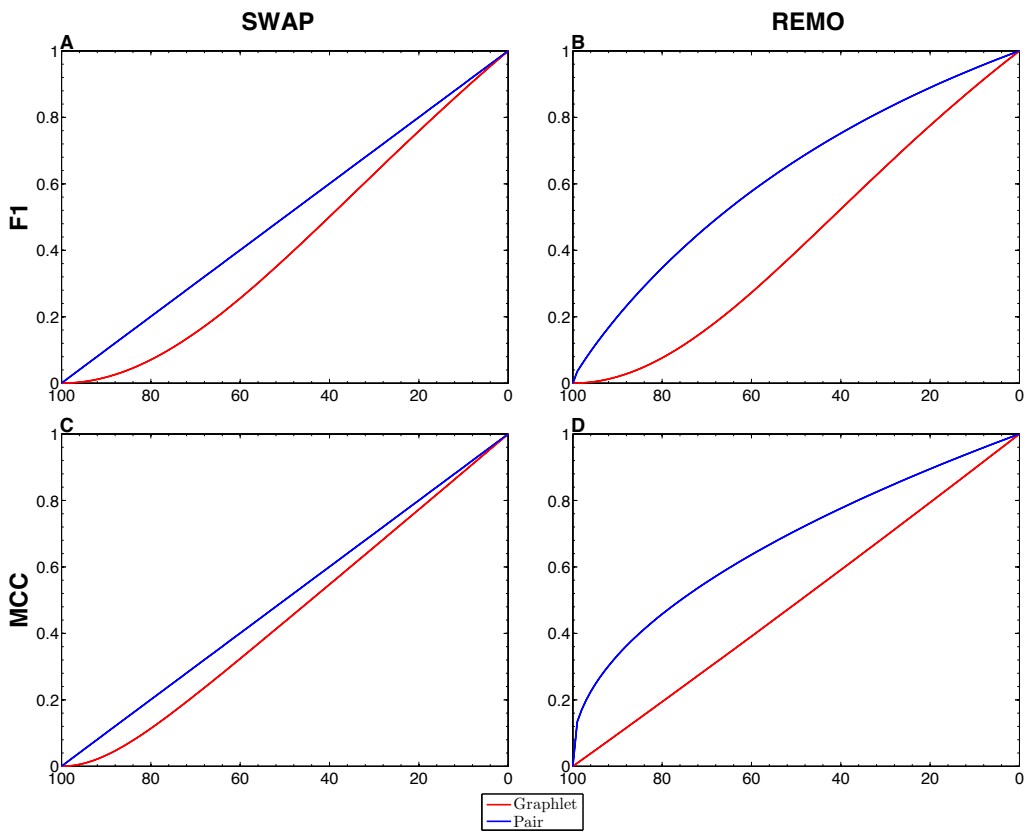

**Figure 2  Comparison between single-edge and GBMs.** For each randomization procedure, average values over $1 \times 10^3$ replicas for single-edge (solid blue line) and graphlet-based (solid red line) F1 and MCC are shown at different percentages of randomization. (A and B) Show F1 for SWAP and REMO randomizations respectively; and (C and D) show MCC for the SWAP and REMO cases respectively.

with Pearson's correlation than with Spearman's rank correlation, where the relationship between Neighborhood Connectivity and GBMs is especially strong.

Concerning the agreement between specific TFs whose local topology varies detected by the difference in centralities and by GBMs, these results are shown as confusion matrices in Table 5. In this case, nodes whose topologies were different in the two compared networks and were detected by differences in centrality and by GBMs are considered TPs; those detected only by a node centrality are FPs; FNs are identified only by GBMs, and those nodes that did not have any variation are TNs. Notably, GBMs are in better agreement with Neighborhood Connectivity, while the larger differences are with Betweenness Centrality. Nevertheless, there are differences in the specific nodes showing variations in all comparisons.

## Subnetwork of *ompR*

Figure 4 depicts the subnetwork formed by all graphlets in which *ompR* participates. This subnetwork is formed by all those nodes that are also part of the graphlets in which *ompR* is one of the nodes and all connections found in these graphlets in any of the two network states. There are 84 TF-coding genes in this network, out of 761 nodes (only four genes are

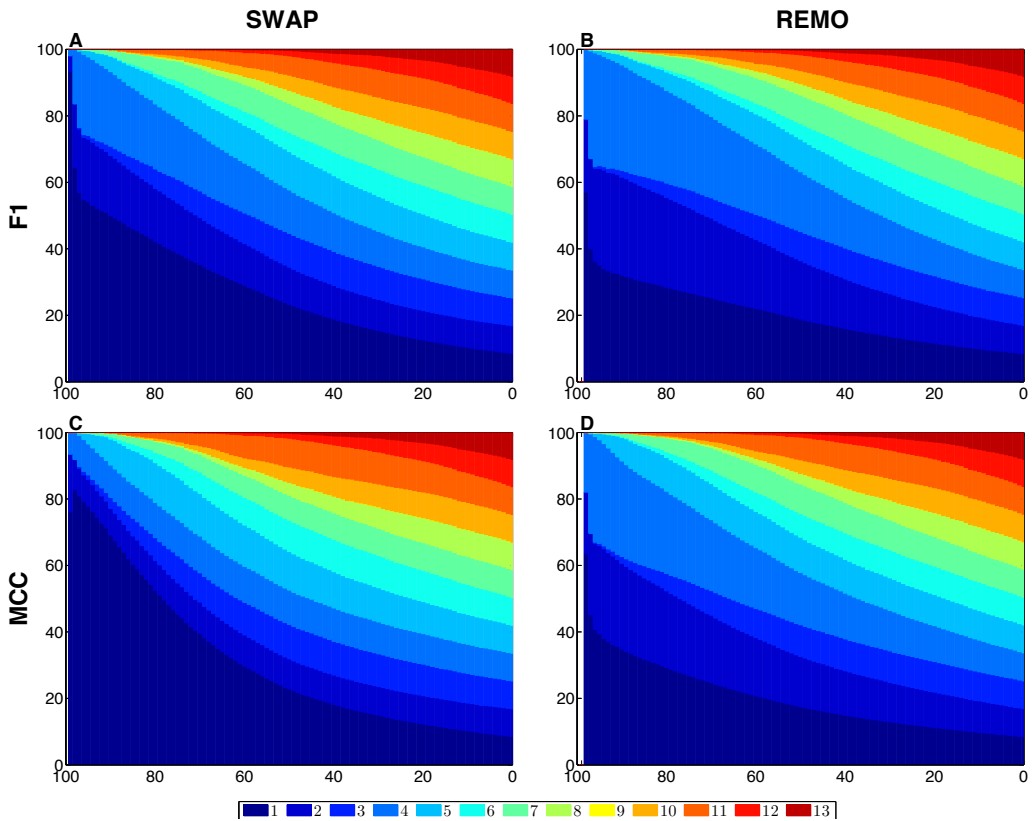

**Figure 3 Contribution of each graphlet type to F1 and MCC GBMs on the two randomization procedures of the *E. coli* reference network.** For each randomization procedure, the plots show the contribution of each graphlet type to the averaged values of each metric over the $1 \times 10^3$ replicas. The *X*-axis indicates the percentage of randomization, ranging from total randomization on the lefthand side to no variation on the right side. The *Y*-axis indicates the contribution of each graphlet type to the metric in the form of a percentage. (A) shows F1 for the SWAP randomization, and (B) F1 for the REMO randomizations respectively; and (C and D), MCC for the SWAP and REMO cases, respectively.

absent in the knock-out state). TF-coding nodes in this subnetwork are connected to their respective target genes by 2,325 edges. Of these regulatory interactions, 31 are present only in the wild-type network (FN edges) and only seven in the state corresponding to the *ompR* knock-out (FP edges). With respect to the subnetwork formed by the direct neighbors of *ompR* (small inset), there are 8 TF-coding genes out of 21 nodes and five edges that are only in the wild-type GRN (FN edges), while 43 connections are present in the two network states (TP edges).

As expected, all direct neighbors of *ompR* are part of this subnetwork, including genes coding for the three sRNAs (OmrA, OmrB and MicF), the genes of the OmpC porin, DtpA, FadL, Sra, NmpC, OmpF and BolA, and the operons *csgDEFG*, *ecnAB* and *flhDC*. According to EcoCyc (*Keseler et al., 2010*), these genes are related to functions that include the formation of curli, the formation of biofilms, the composition of the outer membrane, uptake of small ligands, and the regulation of other genes involved in these functions. Nevertheless, our approach evidenced other differences between the wild

Table 4 **Correlation between differences in node centralities and GBMs for TF-coding genes.** Pearson's (upper right) and Spearman's (lower left) correlations computed between node centralities and GBMs calculated for TF-coding genes on the comparison between the wild type GRNs of *E. coli* and *ompR* knock-out. Centralities metrics are: Average Shortest Path Length (ASPL), Betweenness Centrality (BC), Closeness Centrality (CLC), Clustering Coefficient (CC), Eccentricity (ECC), Neighborhood Connectivity (NC), Stress (STR), Degree (DEG, sum of outdegree and indegree), Outdegree (ODE), and Indegree (IDE). GBMs are F1 and MCC. Statistically significant correlation coefficients (*p*-value ≤ 0.01) are shown in bold and their backgrounds are shaded in gray.

| | ASPL | BC | CLC | CC | ECC | NC | STR | DEG | ODE | IDE | F1 | MCC |
|---|---|---|---|---|---|---|---|---|---|---|---|---|
| ASPL | – | 0.058 | **0.449** | **0.532** | **0.979** | **0.397** | 0.042 | **0.861** | **0.865** | **0.601** | **−0.238** | **−0.217** |
| BC | **0.328** | – | −0.012 | 0.002 | 0.108 | −0.017 | **0.992** | 0.072 | 0.028 | 0.135 | 0.012 | 0.014 |
| CLC | **0.945** | **0.341** | – | **0.566** | **0.360** | **0.595** | −0.018 | **0.678** | **0.605** | **0.616** | −0.176 | −0.174 |
| CC | **0.568** | **0.338** | **0.583** | – | **0.480** | **0.937** | 0.001 | **0.658** | **0.469** | **0.821** | −0.034 | −0.029 |
| ECC | **0.519** | **0.189** | **0.497** | **0.510** | – | **0.331** | 0.092 | **0.830** | **0.832** | **0.582** | **−0.218** | **−0.197** |
| NC | **0.677** | **0.358** | **0.661** | **0.626** | **0.422** | – | −0.022 | **0.593** | **0.408** | **0.765** | −0.003 | −0.001 |
| STR | **0.468** | **0.709** | **0.482** | **0.499** | **0.299** | **0.496** | – | 0.057 | 0.011 | 0.127 | 0.017 | 0.019 |
| DEG | **0.425** | **0.250** | **0.414** | **0.666** | **0.657** | **0.519** | **0.378** | – | **0.948** | **0.805** | −0.184 | −0.173 |
| ODE | **0.391** | 0.109 | **0.388** | **0.533** | **0.760** | **0.427** | 0.197 | **0.775** | – | **0.575** | **−0.235** | **−0.222** |
| IDE | **0.392** | **0.271** | **0.385** | **0.699** | **0.592** | **0.503** | **0.398** | **0.951** | **0.694** | – | −0.034 | −0.030 |
| F1 | **−0.589** | **−0.292** | **−0.567** | **−0.504** | **−0.371** | **−0.821** | **−0.440** | **−0.431** | **−0.319** | **−0.402** | – | **0.999** |
| MCC | **−0.589** | **−0.292** | **−0.567** | **−0.504** | **−0.371** | **−0.821** | **−0.440** | **−0.431** | **−0.319** | **−0.402** | **1.000** | – |

Table 5 **TF-coding nodes identified by centralities and graphlet based F1.** The table shows confusion matrices of TF-coding genes whose variation in local topology was identified by differences in the centrality metrics and by F1 based on graphlets. This table was built on the comparison between GRNs of *E. coli* for wild type and *ompR* knock-out conditions. In this case, nodes identified by both approaches are considered TPs; those whose topological variation was identified only by a change in node centrality are FPs; while those solely identified by F1 are considered FNs. Nodes that do not show any variation in their topology are TNs. Centralities metrics are: Average Shortest Path Length (ASPL), Betweenness Centrality (BC), Closeness Centrality (CLC), Clustering Coefficient (CC), Eccentricity (ECC), Neighborhood Connectivity (NC), Stress (STR), Degree (DEG, sum of outdegree and indegree), Outdegree (ODE), and Indegree (IDE).

| | TP | FP | TN | FN |
|---|---|---|---|---|
| ASPL | 40 | 8 | 131 | 18 |
| BC | 35 | 45 | 94 | 23 |
| CLC | 40 | 8 | 131 | 18 |
| CC | 23 | 1 | 138 | 35 |
| ECC | 11 | 1 | 138 | 47 |
| NC | 51 | 1 | 138 | 6 |
| STR | 30 | 8 | 131 | 28 |
| DEG | 14 | 1 | 138 | 44 |
| IDE | 13 | 1 | 138 | 45 |
| ODE | 8 | 1 | 138 | 50 |

type and the knock-out network. There are several TF-coding genes which are not in the direct neighborhood of *ompR* but are still part of its subnetwork and were present only in one of the two states. Five TF-coding genes are only present in the wild type (*yeiL*, *mlrA*, *feaR*, *rhaR* and *rhaS*) while a fifth TF is only expressed in the knock-out (*tdcA*). *yeiL* encodes for a TF with no known targets but itself; *mlrA* is part of the signaling cascade

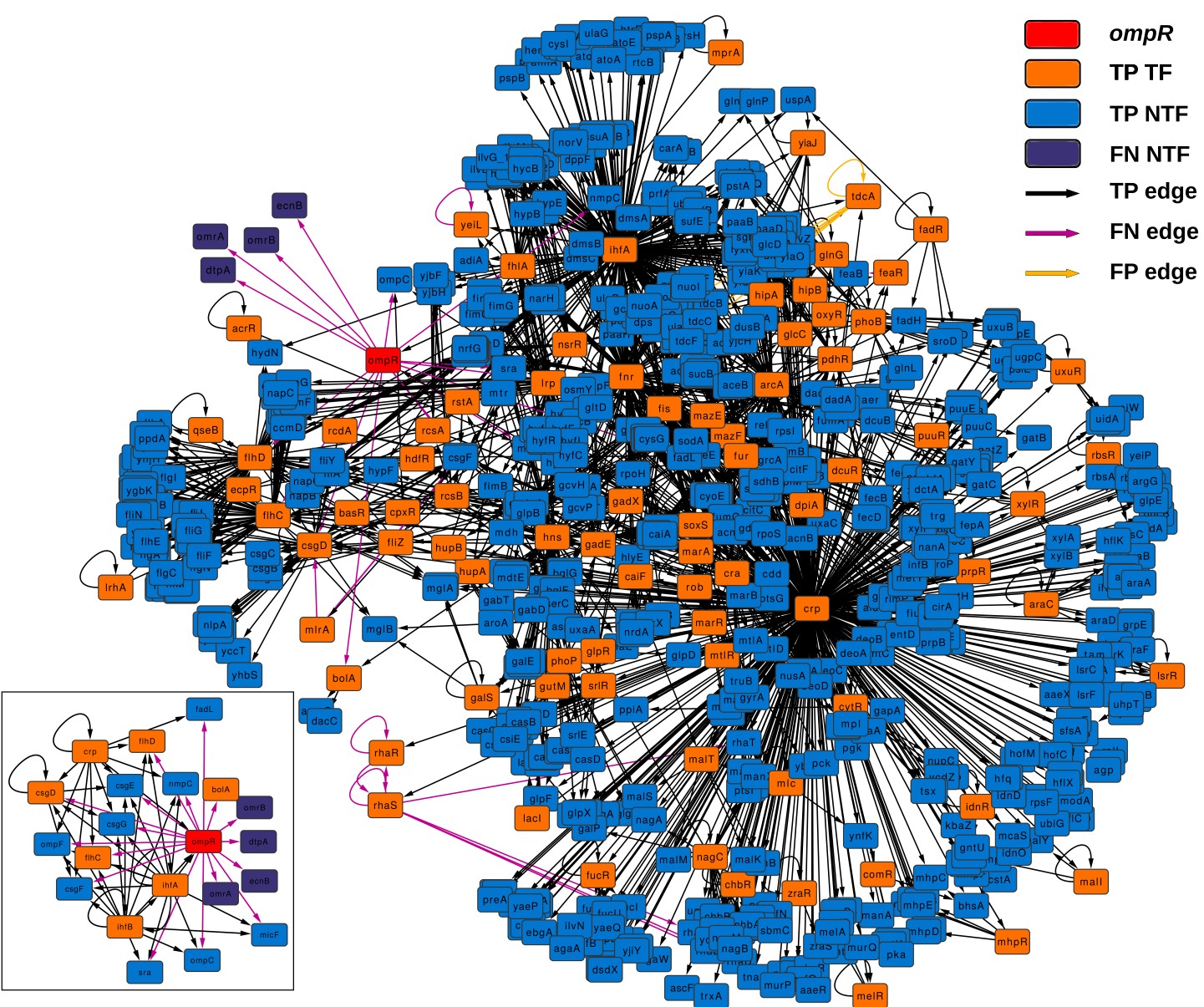

**Figure 4** *ompR* **subnetwork.** Subnetwork formed by all graphlets in which *ompR* participates (red colored node) showing the comparison between wild-type and the *ompR* knock-out GRNs. The subnetwork elements are displayed using different colors for TF-coding genes and effector genes. TP elements are those present in both networks being compared, FN are network elements present only in the wild-type network and FP are those elements present only in the *ompR* network. The small insert represent the subnetwork formed by only direct neighbors of *ompR* in the comparison using the same coloring scheme.

that controls the biosynthesis of curli; *feaR* is considered an activator of phenylacetate synthesis from 2-phenylethylamine and its only two direct regulations are *feaB* and *tynA*; *rhaR* and *rhaS* are part of the same operon and their product regulates genes involved in l-rhamnose degradation and transport. TdcA, the product of *tdcA*, controls the tdc operon that contains genes which products are involved in the transport and metabolism of threonine and serine. With respect to TF-coding genes present only in one state but

are not part of the *ompR* subnetwork, solely two nodes are only present in the wild type network: *tdcR* and *ydeO*. The TFs encoded by *tdcR* and *tdcA* are positive regulators of the *tdc* operon, but in contrast to *tdcA*, which is part of this operon, *tdcR* is not. On the other hand, YdeO induces the expression of genes involved in the response to acid resistance, including respiratory genes and four TF-coding genes governing stress response.

These networks and the results of their comparisons are available in the form of a Cytoscape session provided as supplementary material. This session also contains additional metrics for each node, including other metrics calculated by *LoTo* based on the rate of graphlet reconstruction (*Martin et al., 2016*) and if the expression of each gene was detected in each studied condition.

## DISCUSSION

Quantification of gene expression is a widely used approach to determine the effect of genetic alterations, such as deletions, mutations or even differences between diverse conditions. Nevertheless, this technique reports quantitative differences in gene expression while it disregards the causes of these variations. On the other hand, differential network analysis tries to identify the variations in network topology, and thus, it helps to identify the mechanisms that cause the alterations in gene expression profiles.

*LoTo* is a tool to perform differential network analysis of GRNs that makes explicit use of graphlets. In the definition of graphlets used in *LoTo*, both true and false edges are equally considered. Despite the need for proper bibliographic and experimental support for true edges in GRNs, there is no doubt about their relevance. True edges represent how the products of source genes control the expression of target genes, implying both the direction and the causality of the regulation. Due to their importance, most of the current metrics used to describe and compare networks such as shortest paths and centralities only consider true edges, disregarding false ones. Thus, false edges are commonly considered as less informative or simply ignored. However, false edges depict indispensable elements of the network topology because its existence indicates the absence of the regulation. Therefore, once a false edge has been identified, its removal—i.e., conversion to a true edge—implies the apparition of a new regulatory relationship that may influence gene expression.

Graphlets depict local network topology and their existence or absence is treated in *LoTo* as a binary classification problem. By doing so, several metrics applied in this type of problems can provide a quantification of the topological similarity of two compared networks. Notably, only 11 nodes found in the gold standard created from RegulonDB are not included in any graphlet. Thus, the definition of graphlets employed in *LoTo* includes most of the network components present in the gold standard. Interestingly, graphlets that do not require their three nodes to represent regulator-coding genes (types 1 to 6) are by far more numerous than those graphlets in which all three nodes represent regulator-coding genes (types 7 to 13). This is expected when one considers that the number of regulator-coding genes is less numerous than those coding for other gene products, and therefore graphlets that require more regulators are deemed to be less frequent. Another

trend is that the occurrence of graphlets decreases as both the number of true connections and the number of regulator-coding genes in their composition increases. Since the number of regulator-coding genes is smaller, this tendency is also expected because an increment in the number of true edges would require the presence of more regulator-coding nodes. Moreover, type 9 (a cycle) is completely absent in the three networks analyzed. Whether the lack of type 9 graphlets is due to their absence in real GRNs or due to the incompleteness of the *E. coli* gold standard, is yet to be determined.

There are different levels in which network similarity can be measured. The first level is the global topology, where the overall structure of two networks is compared and their topological similarity reported. *LoTo* reports graphlet occurrence in a similar way to other approaches (*Przulj, Corneil & Jurisica, 2004*; *Sporns & Kötter, 2004*; *Przulj, 2007*; *Koschützki & Schreiber, 2008*; *McDonnell et al., 2014*; *Yaveroğlu et al., 2014*). In addition, *LoTo* also makes use of binary classification metrics calculated for the presence or absence of graphlets to quantify the similarity between two states of a network. F1 and MCC were calculated at different percentages of randomization of the *E. coli* gold standard (Fig. 2) to show how these metrics calculated for the presence or absence of graphlets behave in a controlled environment. In all cases, GBMs are below their single-edge counterparts, indicating that GBMs are more sensitive to the percentage of change in the network than single-edge metrics. Moreover, when the metrics are calculated for graphlets, the removal or swapping of an edge has a greater impact on the metrics than when calculated for single edges. This can be foreseen since the change of a single edge may change the type of several graphlets, thus explaining the lower values observed for GBMs. The increased sensitivity of graphlets based metrics becomes especially relevant when considering SWAP randomization (Fig. 2A and 2C), where the addition of edges (FPs) can create new graphlets. As shown in Fig. 3, the contribution of each type of graphlet to F1 and MCC is sensitive to the percentage of change. This is particularly relevant at high percentages of change, where both metrics F1 and MCC are dominated by simpler graphlets of types 1, 2 and 4. This is expected when considering that the formation of these graphlets require only two true edges and the highest number of false edges among all graphlet types. It is also very important to consider that the arbitrary introduction of true edges in the SWAP randomization increments the occurrence of these simpler graphlets as the percentage of alteration increases, while in REMO, simpler graphlets only appear by decomposition of more complex ones.

The second level of network similarity is local topology. In this case, the goal is to report how well maintained are the relationships of individual genes with the rest of the network. Variations in degree and other measures of node centrality can be used to detect nodes that experience variations in their relationships with other genes, i.e., how their regulatory relationships are altered. For this purpose, *LoTo* calculates the binary classification metrics for the existence or absence of all graphlets in which the same node participates. As an example of this second level of topological similarity, *LoTo* was used to identify TF-coding genes showing differences in their local topology in two condition specific networks. These two GRNs represent *E. coli* wild-type and a knock-out of *ompR*. As evidenced in Table 4, graphlet based F1 and MCC do not show strong correlations

with most of the differences in node centralities. Notably, this indicates that the various metrics and centralities capture diverse aspects of the network topology and thus, each metric depicts diverse traits of variation in the local topology. This is confirmed in Table 5, where it is evident that each metric identifies different TF-coding genes as those whose local topology varies in the compared networks, even though the agreement (TPs +TNs) is larger than the disagreement (FPs +FNs). Interestingly, the main difference between GBMs and the other metrics are due to the explicit usage of graphlets. As shown in Fig. 4, the subnetwork of a gene formed by all graphlets in which that node participates contains a large fraction of the entire network, almost half of it in the example shown. This subnetwork includes not only direct neighbors of a node, but also its neighbors in second grade and the relationship between them. Therefore, the higher similarity of GBMs with Neighborhood Connectivity is expected, since this centrality quantifies links between the direct neighbors of a node. In a similar way, the disagreement between GBMs and Betweenness Centrality is also expected, since it counts the number of shortest paths that traverse a node and thus includes all nodes in the network in its calculation. In relation to the *ompR* subnetwork, six out of eight TF-coding genes that are only present in one of the network states are part of it. This indicates an interconnection between these regulators that is explicitly found by our graphlet based approach. Importantly, the function of genes regulated by these TFs is related to the main functions previously reported in acid stress response (*Stincone et al., 2011*; *Johnson et al., 2014*). These results evidence that the approach followed finds similar results to the more traditional transcriptome profiling, and simultaneously provides the means to identify regulatory relationships that would have been obviated otherwise.

There is a third level in which network topology can be studied: the identification of the individual edges and nodes that disappear or appear in the comparison of two GRNs. Even if this level is not explicitly treated in this work, it is implicitly employed in *LoTo*, as changes in single edges alter graphlet types. Nonetheless, this information is explicitly provided in the output of *LoTo*.

## CONCLUSIONS

Given the results shown, the GBMs calculated by *LoTo* are proposed as novel indicators of the topological similarity between different realizations of the same GRNs. In addition, *LoTo* is able to identify those nodes whose local topology varies in GRNs, and hence, show differences in their regulation. Notably, by using graphlets instead of single edges, the approach implemented in *LoTo* captures topological variations that are not detected by other metrics and would be disregarded otherwise. Our approach can also be used to perform topological comparisons of any type of directed network, as long as different states of those networks are available.

## ACKNOWLEDGEMENTS

The authors would like to thank Dr. Jose Antonio Garate and Sebastian E. Gutierrez-Maldonado for all the useful discussions and an initial revision of the manuscript.

### Funding

This study received partial economical support from Proyecto Financiamiento Basal PIA CONICYT[PFB16] and ICM-Economıa project to Instituto Milenio Centro Interdisciplinario de Neurociencias de Valparaiso (CINV) [P09-022-F]. AJMM and SCR received economical support from FONDECYT Iniciacion project [11140342]. CD received economical support from FONDECYT postdoctoral project [3140007]. Powered@NLHPC: This research was also supported by the supercomputing infrastructure of the Chilean National Laboratory for High Performance Computing (NLHPC) [ECM-02]. There was no additional external funding received for this study. The funders had no role in study design, data collection and analysis, decision to publish, or preparation of the manuscript.

### Grant Disclosures

The following grant information was disclosed by the authors:
Proyecto Financiamiento Basal PIA CONICYT to Fundacion Ciencia & Vida (FCV): PFB16.
ICM-Economıa project to Instituto Milenio Centro Interdisciplinario de Neurociencias de Valparaiso (CINV): P09-022-F.
FONDECYT Iniciacion project: 11140342.
FONDECYT postdoctoral project: 3140007.
Chilean National Laboratory for High Performance Computing (NLHPC): ECM-02.

### Competing Interests

Tomas Perez-Acle is an Academic Editor for PeerJ.

### Author Contributions

- Alberto J. Martin conceived and designed the experiments, performed the experiments, analyzed the data, wrote the paper, prepared figures and/or tables, reviewed drafts of the paper.
- Sebastián Contreras-Riquelme performed the experiments, wrote the paper, reviewed drafts of the paper.
- Calixto Dominguez analyzed the data, wrote the paper, reviewed drafts of the paper.
- Tomas Perez-Acle wrote the paper, reviewed drafts of the paper.

### Data Availability

The raw data has been supplied as a Supplementary File.

### Supplemental Information

Supplemental information for this article can be found online at http://dx.doi.org/10.7717/peerj.3052#supplemental-information.

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
