# Peer review of "LoTo: a graphlet based method for the comparison of local topology between gene regulatory networks"

_PeerJ, doi:10.7717/peerj.3052_

## Round 0.1 · original submission · Major Revisions

The manuscript has been carefully evaluated by three external reviewers. They all found major issues and methodological points that need to be properly addressed and that they are attached below. Please, provide a substantially revised version of the manuscript, paying also attention to discuss your method in light of other available such as FANMOD.

·

Basic reporting

No Comments

Experimental design

You are somewhat inconsistent when describing how your GBMs should be used. In the abstract, you say

“different states of a GRN”
“graphlets in a state of the network are compared to those formed by the same three nodes in another state of the GRN”
“LoTo provides a tool to recognize those genes whose network topology has changed between different realizations of a GRN”

You use similar language throughout the text. This language implies a degree of symmetry between the “network states” or “network realizations,” meaning that the two networks in question are somehow equally valid.

However, using language like “false positive” and “false negative” implies that one of the networks is “correct” and the other is some sort of approximation or noisy version of the correct network. Mathematically, there is nothing wrong with arbitrarily labeling one of the networks to be “correct,” since Equations 3 and 4 are both symmetric under switching FN<--->FP. But if you’re claiming that the networks are equally valid, then TP and TN are really measures of “agreement,” and FP and FN are really measures of “disagreement” between the two states. This is fine, but it needs to be explained.
* * *
I do not understand this sentence:

“For completeness, the contribution of each type of graphlet to both metrics at different percentages of change is shown in Fig. 3.”

nor do I understand the caption and plots in Figure 3. What, exactly, are the X and Y axes, and what are the units? And for a given X value and a given Y value, what does the color (Z value) at that point mean? I'm having a difficult time deciphering this from your caption. These plots could be useful, but I cannot tell what you're trying to communicate with these plots and the caption in their current form.
* * *
Under “Characterization of condition specific GRNs,” you are discussing the two subnetworks, wild-type and knockout E. coli. Under the section “Comparison of GBMs with single-edge based metrics” and in Figure 2, it is unclear which network you are randomizing. You eventually clarify in the “Discussion” section:

“F1 and MCC were calculated at different percentages of randomization of the E. coli gold standard (Fig 2) to show how…”

but you should clearly state that you’re changing your focus back to the gold network in the “Comparison of GBMs with single-edge based metrics” section.
* * *
I do not understand the section “Comparison of GBMs with differences in node centralities: identification of nodes with variation in their local topology.” What, precisely, are you correlating? It sounds like the centrality measures you’re using are single-node centrality measures (e.g. the betweenness centrality of gene 1 is X, the centrality of gene 2 is Y, etc.). But as far as I can tell, the GBMs you introduced, F1 and MCC, are measures used to compare one entire network to another entire network, giving no information about individual nodes in an individual network. Pearson and Spearman correlation coefficients have to relate vector A to vector B, both of which must have the same dimension, so I don’t understand how you can compare single-node measures to a measure comparing two networks. Are you using the *average* (mean) centrality of all nodes in the network, and then comparing across different randomizations? If that’s the case, then I guess you’re correlating

(measure A in randomization i) – (measure A in gold network)

with

(measure B in randomization i) – (measure B in gold network)

across many i’s. However, in your “Discussion” section, you say

“…LoTo calculates the binary classification metrics for the existence or absence of all graphlets in which the same node participates.”

which leads me to believe you separately formulated single-node versions of your GBMs. But you never really explain how these are calculated. Maybe I can guess how you did it, but I shouldn't have to guess.

Validity of the findings

The findings are unclear at the moment, so I cannot judge their validity yet.

Additional comments

There are a lot of good ideas in this paper, but you need to be more explicit when explaining what you did. I think your paper would be much easier to understand if you used more equations. For example, you explained how F1 and MCC are computed when comparing two networks, but how exactly do you compute the single-node F1 and MCC?

Minor note: in your introduction, you say "estates," but I think you meant "states."

·

Basic reporting

No comments

Experimental design

No comments

Validity of the findings

- The authors could also take “edge betweenness” into account.
- A discussion of the observed differences between the two E.coli experiments is missing. Such a discussion is needed to emphasize that the topological differences are also biologically relevant.The authors should also discuss where changes in local topology originate that are not in the neighborhood of the knockout gene,
- My main issue with this paper is that it is not evident to me how the network randomization is supposed to demonstrate how GBMs are superior to single-edge methods. The authors fail to motivate this in the introduction and also don’t sufficiently discuss the results (mostly Figure 2). Please expand significantly on this, since it is obviously an important message of the paper.
Similarly, what is the consequence of the results shown in Figure 3? Does this mean that only graphlets of type > 10 are relevant?

Additional comments

Martin et al. suggest a method for studying changes in gene regulation between two experiments using differences in the topology of the two corresponding gene regulatory networks. Instead of focusing on global topology changes, they are interested in the local topology and assess this using graphlets. The authors compare their method to simpler methods such as network centrality and show that considering triplets of nodes in a gene-regulatory-network is a more powerful approach since it considers triplets and thus captures a wider range of topological changes. As a twist, they consider the absence of an edge as equally important to the existence of an edge. To reduce the complexity of the problem, the authors consider not a whole gene interaction but a smaller (sparse) gene regulation network in which transcription factors and their regulated targets are connected. The authors describe a method to apply graphlets in local topology comparison and thus deliver a promising tool for network biology. The manuscript should however be improved to present the results more clearly.

Minor comments:
I highlighted some minor language issues in the attached PDF. In addition, the following points should be addressed.

L. 150: Example of E.coli RNA profiles. I suggest you only mention the two conditions you consider for validation and describe the rationale behind the experiment, e.g. what is the function of EvgS and why is it relevant? Same for compR.
L.167: I do not follow why k is at most an order of magnitude smaller than n. Would that not make it O(t n^2) worst case runtime?
L. 170: Describe the input format for the web server
L. 172: ‘In this case’: Should it not be up to the user to decide whether to binary or continuous numbers for the edges? Why should it be a likelihood? Rather call it ‘score or p-value’ because the choice is the user’s.
L. 178: What is the rate of graphlet reconstruction? Explain in a bit more detail. Include reference to the authors recent PLOS ONE publication?
L. 191: It is not really surprising that the network is sparse. Replace notably with something as ‘As expected’.
L. 201: As happens with network components… I don’t understand this sentence.
Table 4 should be color coded to make differences more easily visible.

Reviewer 3 ·

Basic reporting

The manuscript entitled: "LOTO: A graphlet based method for the comparison of local topology between gene regulatory networks" by A.J.M. Martin et al describes a web server to compare two directed networks based on occurrences of 3-node subgraphs across the two directed networks. The group of Uri Alon has extensively studied the occurrence of 3-node sub-graphs in gene regulatory networks since 2002 (Shen-Orr et al, 2002, Milo et al 2002). Uri Alon and several others have extensively studied the over-representation of sub-graphs or motifs in several real networks. In this manuscript, the authors introduce an alternate term "graphlet" for the "3-node sub-graphs", and have tried to use this new terminology to present known, published and insignificant results as new results. In short, in my view, LOTO is a webserver to compute occurrences of 3-node sub-graphs in directed networks. For example, motif finder and FANMOD (http://theinf1.informatik.uni-jena.de/motifs/) are other tools or web servers which can enumerate 3-node and 4-node sub-graphs in directed networks. I do not see the utility of publishing LOTO a decade after publication of FANMOD. Hence, I do not think the manuscript should be published in PeerJ.

Beyond the primary objection mentioned above, the method described in the present manuscript does not provide any insights into the dynamics of gene regulatory networks. For example, there are hardly any differences in the occurrence of 3-node sub-graphs across the 3 considered networks by authors, i.e., reference, wild-type and ompR. The authors should clearly demonstrate how LOTO can identify key nodes or modules across two networks which cannot be identified based on edge-based or node-based measures.

Another serious issue with this manuscript is that the authors try to not give appropriate credit to previous work in the area. This is reflected through their citation of earlier work in Introduction and other sections.

Experimental design

I do not believe the manuscript in present form describes original results. My comments in the above section should highlight the serious issues with this manuscript.

Validity of the findings

No Comments

Additional comments

No Comments

---

## Round 0.2 · Minor Revisions

The manuscript is remarkably improved and only needs minor remaining revision.

One of the reviewers pointed out some minor corrections for the text that need to be taken into account and these are more than can be accommodated in our production phase, hence you can take on board those edits as appropriate and revise and resubmit.

·

Basic reporting

No comment

Experimental design

No comment

Validity of the findings

No comment

Additional comments

No comment

·

Basic reporting

The English in the manuscript has improved a lot. Nevertheless, there are still minor issues to be fixed. I have attached a document highlighting small issues in the Introduction.

Experimental design

no comment

Validity of the findings

no comment

Additional comments

The authors have significantly improved their manuscript and addressed all of my previous comments satisfactorily.

---

## Round 0.3 · accepted · Accept

I am really glad to endorse this nice article for publication on PeerJ.